# An Event-Driven Self-Clocked Digital Low-Dropout Regulator with Adaptive Frequency Control

Yen-Ming Chen and Ching-Jan Chen *

Department of Electrical Engineering, National Taiwan University, No. 1, Sec. 4, Roosevelt Rd., Taipei 10617, Taiwan; f07921027@ntu.edu.tw
* Correspondence: chenjim@ntu.edu.tw; Tel.: +886-2-3366-3366 (ext. 348)

**Abstract:** Digital low-dropout (DLDO) is widely used for power management in the system-on-chip (SoC) because of its low-voltage operation and process scalability. However, conventional DLDOs suffer from the trade-off between transient response and power consumption of the DLDO and the clock generator. This paper proposes an event-driven self-clocked DLDO regulator. The proposed low quiescent current ($I_Q$) event-driven adaptive frequency clock generator (EACG) adapts its frequency in different load conditions without a current sensor or complex compensation circuit for stable operation in the entire load range. The proposed DLDO does not need any external clocking signal and can maintain low output ripple and low power consumption in the steady-state. The clock-less transient detector (CLTD), consisting of two clock-independent transient detection paths, uses power more efficiently and improves the transient response significantly without sacrificing the power consumption. This work was fabricated in a 40 nm CMOS process with an 0.3 nF on-chip capacitor. The measurement results show that with the step load current between 1 mA and 60 mA, the proposed DLDO achieves a transient recovery time of 220 ns. The total $I_Q$ of the proposed DLDO is only 26 μA in steady-state, and it achieves stable operation in the entire load range.

**Keywords:** digital low-dropout (DLDO) regulator; event-driven; low power consumption; adaptive frequency; system-on-chip (SoC)





## 1. Introduction

Modern system-on-chip (SoC) designs employ low-dropout regulators to generate multiple distributed power domains for different sub-circuits, such as radio frequency (RF) circuits, analog circuits, and digital circuits. Each of them has its own operating power level, and the features of the supply voltages are quite different, too. Analog low-dropout (ALDO) regulators have been widely used for such demands for a long time because of their advantage of high power density [1–5]. The ALDOs have mature developments and great characteristics such as low quiescent current [1], high power supply rejection ratio [2,3], and fast transient response [4,5]. However, when the supply voltage (VDD) becomes less than 1 V, the ALDOs' performance faces several difficulties due to degrading voltage gain and greater susceptibility to the process-voltage-temperature (PVT) variations. Instead, digital low-dropout (DLDO) regulators have become valued due to their voltage scalability feature. Recently, digital low-dropout (DLDO) regulators have been widely used for low input voltage power processing in SoC [6–23]. With the process advances and the growing demand for portable devices, digital circuits' supply voltages (VDDs) are now lower.

The DLDOs can be classified into synchronous DLDOs [6–14] and asynchronous DLDOs [15–23]. Synchronous DLDOs use a constantly toggling synchronous clock signal to drive the comparator and the digital controller. Therefore, the control loop latency and the droop response of the synchronous system are highly dependent on the sampling frequency $F_{CLK}$. Because the dynamic comparator (or the clock-triggered analog-to-digital

converter (ADC)) and the controller switch at the edge of the clock signal, the response time ($T_{RES}$) is around 1~1.5 clock cycles for a synchronous DLDO to respond to the load transient. From the capacitor charging equation, we can obtain the output voltage drop:

$$V_{DROP} = \frac{\Delta I_{LOAD} \cdot T_{CLK}}{C_{OUT}} \tag{1}$$

where $\Delta I_{LOAD}$ is the difference in load current, $T_{CLK}$ is the clock cycle time, and $C_{OUT}$ is the output capacitance. Therefore, to reduce the voltage drop caused by a severe load current change, the synchronous DLDOs have to use a high-frequency clock or a large output decoupling capacitor. However, a fast clock inevitably results in high power dissipation, and a large on-chip capacitor occupies a large area and leads to higher costs. Thus, synchronous DLDOs intrinsically suffer from the bottleneck of clock frequency-dependent delay.

Asynchronous DLDOs have gained great attention in recent years since they break the limitation of synchronous DLDOs. In [15–18], they change the digital controller output directly to avoid the bottleneck of clock speed. The design in [19,20] uses a continuous-time ADC to detect the output voltage ($V_{OUT}$) deviations. However, using a pipeline control loop working with a synchronous controller still takes a long recovery time ($T_R$) unless using a fast clock signal. Using continuous-time ADC makes the architecture more complex and sensitive to input voltage variation. Furthermore, using a high-speed continuous-time ADC or a high-speed comparator to detect load transient consumes huge power to maintain a fast response since it has to sense the actual voltage level. Many asynchronous DLDOs require high-frequency external clocks. These high-frequency voltage-controlled oscillators (VCO) consume huge amounts of power and are sometimes even larger than the power consumption of a DLDO.

Another common issue for DLDOs is the stability problem at light-load conditions. Not only does a DLDOs' output pole frequency vary with the load current, but the stability criteria are also related to the sampling frequency of the digital circuits. To avoid this problem, some prior research has used a fixed sampling frequency designed for the minimum load current condition [6]. However, this will make the recovery time longer and limit the loading range. For pursuing better performance, the fixed frequency DLDO designs in recent years mostly add a complex digital compensator to avoid unstable operation. In [14], they analyzed the limit cycle oscillation (LCO) phenomenon of conventional fixed-frequency DLDO. They present a solution by adding auxiliary power transistors to add a zero at the output node. In addition, most of the capacitor-less DLDOs ignored the parasitic capacitance at the output node, which is often quite large in a Very Large-Scale Integration (VLSI) system [18]. When the parasitic capacitance at the supply node of enormous digital circuits is up to dozens of pF levels, it likely will affect the stability of a high-frequency clock capacitor-less DLDO.

In this paper, an event-driven self-clocked DLDO regulator is proposed to solve the aforementioned issues of synchronous and asynchronous DLDOs. A clock-less transient detector (CLTD) and an event-driven adaptive frequency clock generator (EACG) are presented in the proposed DLDO. Furthermore, EACG adapts its frequency to load current without a current sensor. Thus, stable operation in the entire load range is achieved with a simple circuit. We separate the detect path into the fast path and slow path, where one can provide a fast response, and the other can ensure the accuracy of the $V_{OUT}$. Since the fast path only has to transfer the output voltage swing into a pulse signal to activate the switching controller, a fast load transient response with a low quiescent current can be achieved. The slow transient detection path ensures a smaller DC error. This paper is organized as follows. In Section 2, we introduce the architecture and the operation of the proposed DLDO and give some introductions to its stability. Section 3 presents the detailed circuit implementation and analysis of this work. The measurement results are shown in Section 4. Finally, we give a conclusion in Section 5.

## 2. The Concept of the Proposed DLDO

The simplified block diagram of the proposed event-driven self-clocked DLDO is shown in Figure 1. There are five main parts in the system, including the clock-less transient detector (CLTD), the event-driven adaptive frequency clock generator (EACG), the digital switching controller (DSC), the binary-weighted pMOS power transistor array, and a dynamic comparator. During the transient period, CLTD distinguishes the $V_{OUT}$ slope (sharp/smooth) and direction (under-/over-shoot), and EACG adapts its clock signal (CLK) frequency based on turn-on bits SW<10:0>. The TD signal is the "transient detected signal" generated by the CLTD to trigger the system at the beginning of the transient event. The TF signal is the "transient finished signal" generated by the DSC to shut down EACG, the comparator, and DSC and reset the CLTD at the end of the transient event.

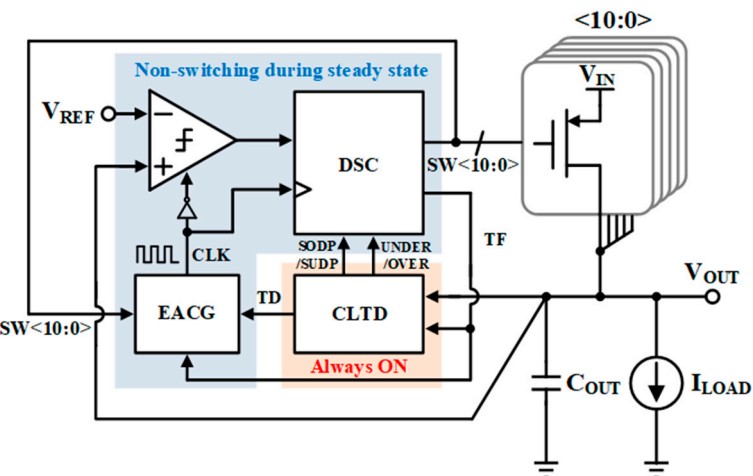

**Figure 1.** Simplified block diagram of the proposed DLDO.

### 2.1. Steady-State Operation

When the system is in the steady-state (between two transient events), only the CLTD will remain operational. Because the EACG will not generate the clock signal at this time, the DSC and the dynamic comparator will not switch and will consume little power. The total quiescent current of the system comes from the CLTD and leakage current. Since the power transistor array will not switch, no ripple will occur at the $V_{OUT}$ node. This can make the output voltage cleaner.

### 2.2. Transient Period Operation

As Figure 1 shows, when a load transient (event) occurs, the CLTD tells the DSC which direction (under or over) and what kind (sharp or smooth) the load transient is and sends the TD signal to trigger the EACG simultaneously. After the EACG receives the TD signal, it generates the clock signal immediately and adapts the sampling frequency based on the present load current ($I_{Load}$) to ensure stable operation. Due to the feature of DLDOs, in that the power transistor's turn-on bits represent the output current, this work achieves load adaptive clock frequency by feedbacking SW signal (turn-on bits) instead of using complicated load current sensing methods. The adaptive frequency control method details will be introduced in Section 3.

The proposed event-driven DLDO can significantly reduce the $V_{OUT}$ drop compared to traditional DLDOs, which use a clock-related detecting technique. Figure 2 shows the load transient response of the synchronous DLDO and the proposed DLDO, where $t_{delay}$ is the delay of CLTD. It can be seen that the first positive edge of the clock will always follow closely to the load current change for the proposed DLDO. Therefore, the system's response time will no longer be related to sampling frequency.

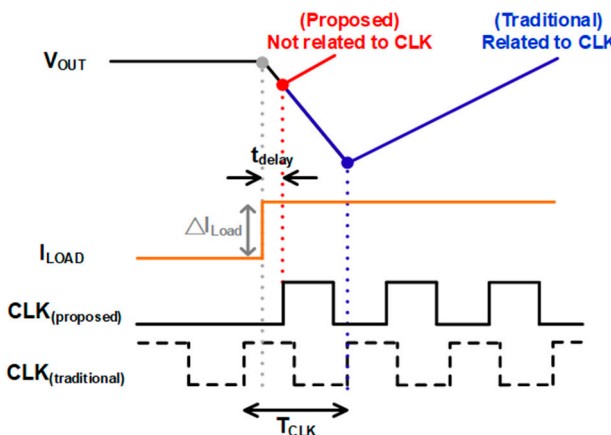

**Figure 2.** Load transient response of the synchronous DLDO and the proposed DLDO.

After the EACG starts generating clock signals, the DSC will start to search the correct turn-on bits of the pMOS power transistor array according to the load transient features, as shown in Table 1.

**Table 1.** The searching method of DSC with different CLTD outputs and different types of $V_{OUT}$ transient.

| Fast Path Output | Slow Path Output | MODE | DSC Reaction |
|---|---|---|---|
| 1 | X | 1 (sharp) | **Binary search**<br>① $V_{OUT}$ drops:<br>Turn on $\overline{MSB}$<br>at the first clock edge<br>② $V_{OUT}$ rises:<br>Turn off $\overline{MSB}$<br>at the first clock edge |
| 0 | 1 | 0 (smooth) | **Linear search** |
| 0 | 0 | 0 (unchanged) | **Steady-state** |

Once the $V_{OUT}$ slope rises/falls sharply ($|dV_{OUT}/dt|$ is large), the signal "MODE" in the DSC will pull high, which means that a severe load transient occurred. Therefore, the DSC will begin the binary search when the CLK starts toggling. The operation of the DSC will depend on the $V_{OUT}$ direction told by the CLTD. If the $V_{OUT}$ rises sharply, the DSC will turn on the most significant bit (MSB) of the power transistors array at the first positive edge of the clock. Then, at every negative edge of the clock, the dynamic comparator will compare the $V_{OUT}$ and the reference voltage ($V_{REF}$), and send the result to the DSC. The DSC will switch the [MSB-1] bit at the next positive clock edge and do this cyclically to the least significant bit (LSB).

When the binary search is over, the DSC will turn to use the linear search to ensure the $V_{OUT}$ is back to $V_{REF}$. If the output of the comparator is three consecutive opposite values, the DSC will determine that the system is back to steady-state and send out the TF signal. The signal TF will shut down the EACG and reset the CLTD. Finally, the system will stop switching and wait for the next event.

If the TD goes high while MODE is still 0, it means there is a slight load transient occurring or the $V_{OUT}$ deviates outside the design window. When the clock starts toggling, the DSC will carry out the linear search directly to prevent unnecessary huge $V_{OUT}$ swings caused by the binary search. Similarly, if the output of the comparator is three consecutive opposite values, the DSC will send out the TF signal to shut the EACG down and reset the CLTD, too. The transient behaviors of MODE = 1 and 0 are shown in Figure 3.

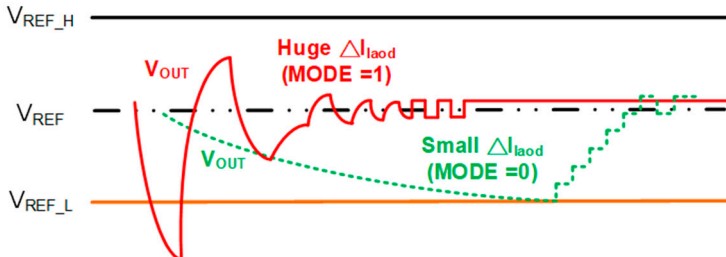

**Figure 3.** Transient behaviors of MODE = 1 and 0.

Since the proposed DLDO separates the transient detector into two parts, it achieves a fast transient response and maintains low power consumption in the steady-state. The fast paths only have to detect the severe $V_{OUT}$ swings instead of obtaining the actual $V_{OUT}$ level. Therefore, the detecting circuits can utilize the power more efficiently and do not need a high-speed but power-hungry detecting circuit. Meanwhile, the slow paths ensure the $V_{OUT}$ will not gradually deviate from the $V_{REF}$. This deviation may happen when there is a slow load transient event, or the load current does not exactly match the discrete steps of current being provided by the pass transistors array. Thus, the DLDO can turn off its clock during steady-state but still ensure a small DC error. The window consisting of $V_{REF\_H}$ and $V_{REF\_L}$ can clamp the $V_{OUT}$ in an acceptable DC deviation range. Most importantly, both the fast paths and the slow paths can be realized by simple low-quiescent current circuits.

### 2.3. Stability

In [13,19], the stability of common fixed sampling frequency DLDOs with PI controllers was analyzed in detail. The linear small-signal ac model of a binary search DLDO in the z-domain is derived in [13], as shown in Figure 4a. The loop gain of such a second-order feedback control loop is given by

$$G(z) = \frac{K(i) \cdot (1 - z_L) \cdot z}{(z - 1) \cdot (z - z_L)} \tag{2}$$

where $K(i)$ is the gain, and the loop gain has two poles: the integrator pole ($z = 1$) and the output pole ($z_L = e^{-f_L/f_{CLK}}$). The corresponding continuous-time loop gain transfer function $G(s)$ can be given by

$$G(s) = \frac{\omega_n^2}{s \cdot (s + 2\eta\omega_n)} \tag{3}$$

The loop gain $G(s)$ contains two poles, where $z = 1$ and $z_L$ in the z-domain map to $s = 0$ and $f_L$ in the s-domain. The Bode diagram of $G(s)$ is shown in Figure 4b.

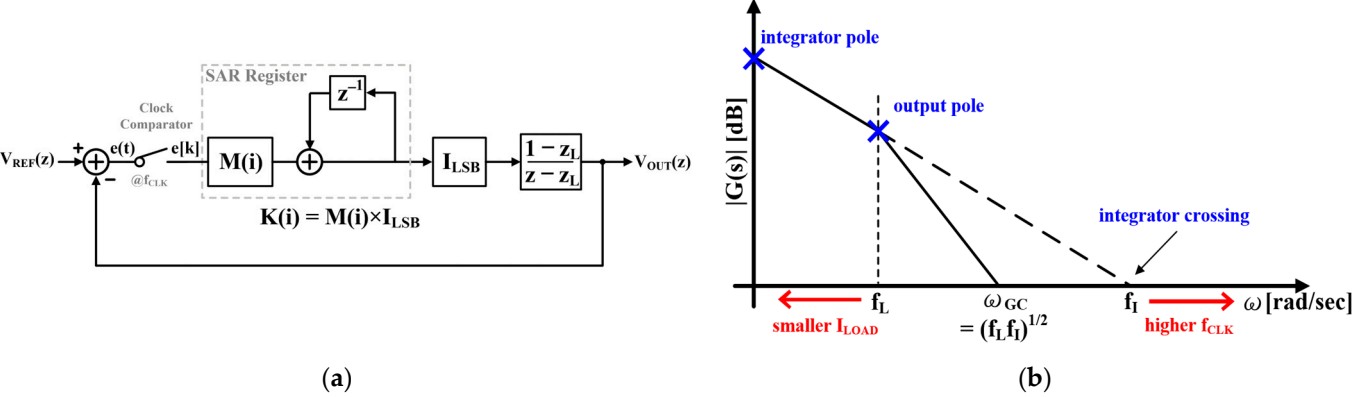

(**a**)

(**b**)

**Figure 4.** (**a**) Small-signal ac model of the binary search DLDO. (**b**) Bode diagram of DLDO loop gain.

The phase margin of loop gain is determined by $f_L$ and $f_I$, where $f_L$ is inversely proportional to the output load RC-time constant and $f_I$ is proportional to the sampling frequency $f_{CLK}$. The phase margin of G(s) is derived as (4) [13].

$$PM = 90° - \tan^{-1}(\frac{f_I}{f_L})^{1/2} \tag{4}$$

Increasing the sampling frequency can lead to a wider bandwidth and a faster response for a given output load. However, the phase margin is reduced at light-load conditions, causing an instability problem. In [13,19], they insert a proportional derivative (PD) controller or compensator into the loop to ensure the stability of the DLDO.

This paper proposes an event-driven adaptive frequency clock generator (EACG) to tackle this issue. The EACG can adapt its frequency to be proportional to the $I_{LOAD}$, making $f_I$ track $f_L$. According to the previous analysis, the phase margin can be kept at a fixed value at various loading conditions. Thus, the system can operate stably. Instead of adding a current sensor to sense the load information, this paper proposes a method to estimate load information from power transistors' turn-on bits SW<10:0> as shown in Figure 1. This method is an easy and low quiescent power way to solve instability issue without using complex circuits. The design detail will be presented in Section 3.3.

## 3. Circuit Implementation

The detailed architecture of the proposed DLDO is shown in Figure 5. The control circuits include three main parts: the DSC, the CLTD, and the EACG. The proposed CLTD can distinguish which direction (under or over) and what kind (sharp or smooth) the load transient is and wakes the system up. The proposed EACG is only active in the transient period to save power consumption, and it can adapt the clock frequency to achieve a stable operation. The DSC consists of binary/linear searching circuits, an SR-latch, and an OR gate. These circuits control the whole system and switch the pMOS power transistor array to maintain $V_{OUT}$ at the required level. This section will give detailed introductions to the three individual sub-circuits.

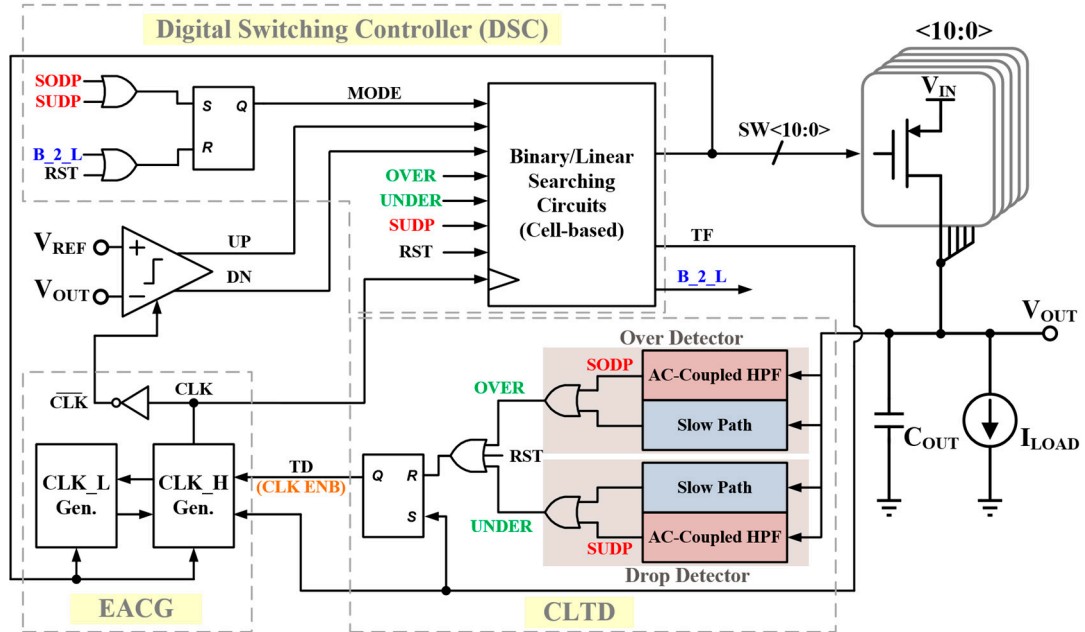

**Figure 5.** Architecture of the proposed DLDO.

### 3.1. Control Logic of the Digital Switching Controller (DSC)

The control logic of the DSC is shown in Figure 6. Once the TD goes high, it means that there occurs a load transient, and the clock will start toggling and make the DSC start operating. The DSC will choose the proper searching method base on the "MODE" signal, as shown in Table 1. Once the $V_{OUT}$ slope rises/falls sharply ($|dV_{OUT}/dt|$ is large, huge $\Delta I_{LOAD}/\Delta t$), the signal "MODE" will be logic-high (=1). Relatively, if the $V_{OUT}$ slope rises/falls smoothly ($|dV_{OUT}/dt|$ is small, less $\Delta I_{LOAD}/\Delta t$), the signal "MODE" will be logic-low (=0).

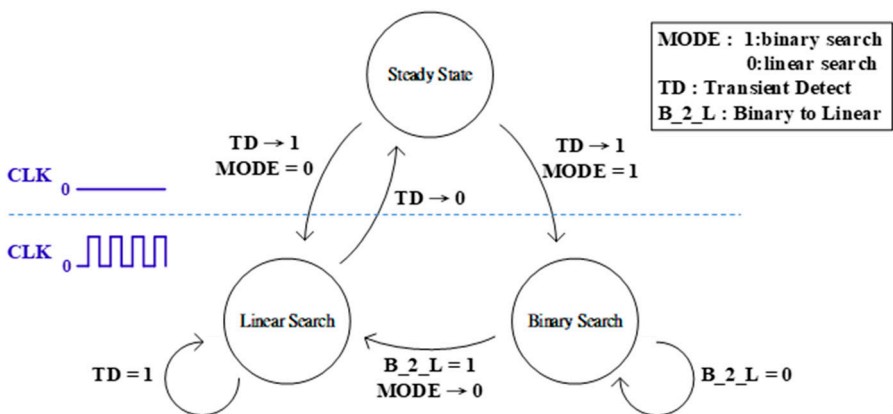

**Figure 6.** State diagram of the proposed DLDO.

As in Figure 5, once the "severe over-shoot detected pulse" (SODP) or the "severe under-shoot detected pulse" (SUDP) pulls high, which means there is a severe load (large $|dV_{OUT}/dt|$), the signal MODE will be pulled to 1, and the TD signal will also be pulled up. Therefore, the DSC will start doing the binary search to recover the $V_{OUT}$ immediately when the CLK starts toggling. Until the binary search is over (search to LSB), the DSC will send out a signal "B_2_L" to reset the SR-latch, and the MODE will be set to 0. Thus, the DSC will carry out the linear search until the system is back to steady-state. Finally, the DSC will send out a pulse "TF" to shut the EACG down and reset the CLTD.

On the contrary, if TD pulls high but MODE still =0, that means the $V_{OUT}$ is out of the clamping window, but the $V_{OUT}$ slope is not sharp enough to activate the ac-couple HPF path. There occurs a slight load (small $|dV_{OUT}/dt|$) transient, and the DSC will start carrying out the linear search to correct the output current without causing huge VOUT swings. Until the system is back to steady-state, the TF pulls high, the TD pulls low, the CLTD is reset, and the clock stops toggling.

Figure 7 shows the timing diagram of an overall operating waveform of the proposed DLDO, taking two load transient events as examples. Assume that there comes a severe $I_{LOAD}$ step-up load transient event (large $dI_{LOAD}/dt$, $dI_{LOAD}/dt > 0$, large $|dV_{OUT}/dt|$) first and then a slight $I_{LOAD}$ step-down event occurs (small $dI_{LOAD}/dt$, $dI_{LOAD}/dt < 0$, small $|dV_{OUT}/dt|$) later.

In the steady-state, all the signals remain 0. When the first load transient occurs, SUDP goes high and pulls up TD. Then, TD triggers the CLK signal, and the system starts using binary search to raise $V_{OUT}$ immediately. After the binary search is over, the DSC sends a B_2_L pulse, and the system will turn to use a linear search to make sure $V_{OUT}$ is back to $V_{REF}$ correctly.

In slight load transient conditions, the system will react after $V_{OUT}$ is out of the windows consisting of $V_{REF\_H}$ and $V_{REF\_L}$. The OVER signal pulls up when $V_{OUT}$ exceeds $V_{REF\_H}$ and activates the system. Then the DSC will use a linear search to regulate the $V_{OUT}$ back. The MODE remains 0 from the beginning to the end. After $V_{OUT}$ equals $V_{REF}$, TF will pull high and shut the system down.

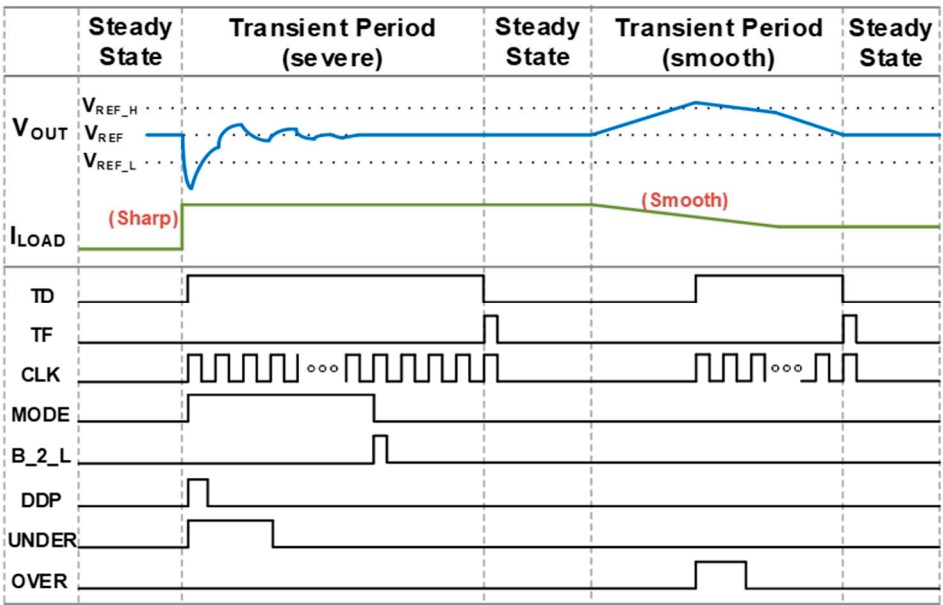

**Figure 7.** The timing diagram of an overall operating waveform.

*3.2. Clock-Less Transient Detector (CLTD)*

The proposed clock-less transient detector (CLTD) is shown in Figure 8. The CLTD consists of two blocks, one is for $V_{OUT}$ over-shoot detection, and the other is for $V_{OUT}$ under-shoot detection. Both of them have a fast signal path and a slow signal path, and this means the CLTD can distinguish which type of load transient occurred and let the digital switching controller (DSC) select the appropriate searching method.

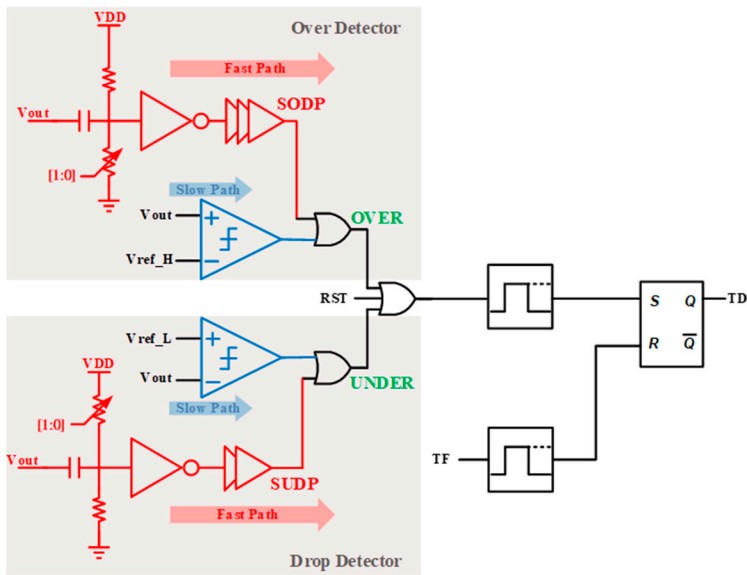

**Figure 8.** The architecture of the proposed CLTD.

The fast signal paths are the $V_{OUT}$ drop detector and the $V_{OUT}$ over detector. They are two ac-coupled pulse generators of a high-pass capacitor, a resistor voltage divider, and an inverter chain. The fast signal paths design is inspired by [18], which demonstrated an efficient drop-detecting method. Thus, similar undershoot performance can be achieved. These simple circuits can transfer a voltage swing to a pulse signal in a short time. Since low VDD conditions or the trigger points of the inverters varied by the process will degenerate the noise immunity, we added two trimming bits to adjust the resistance voltage divider.

These trimming bits can adjust the noise margin of the high-pass paths depending on the different operating situations.

The two slow paths are two 2.5 µA-quiescent-current continuous-time comparators, which compare $V_{OUT}$ with $V_{REF\_H}$ and $V_{REF\_L}$, respectively. $V_{REF\_H}$ and $V_{REF\_L}$ determine the upper-bound and lower-bound of steady-state $V_{OUT}$ range. Once $V_{OUT}$ deviates out of the window, the TD will pull high to trigger the whole system.

With the fast paths and the slow paths in the CLTD, the proposed DLDO can distinguish different load transient situations ($V_{OUT}$ rises/drops, large/small $|dV_{OUT}/dt|$). Hence, the DSC can choose the proper searching method to regulate the output voltage, as shown in Table 1.

### 3.3. Event-Driven Adaptive Frequency Clock Generator (EACG)

The proposed event-driven adaptive frequency clock generator (EACG) is shown in Figure 9. The EACG consists of two similar clock generator sub-circuits, and they control the logic-high time and logic-low time of the clock signal, respectively.

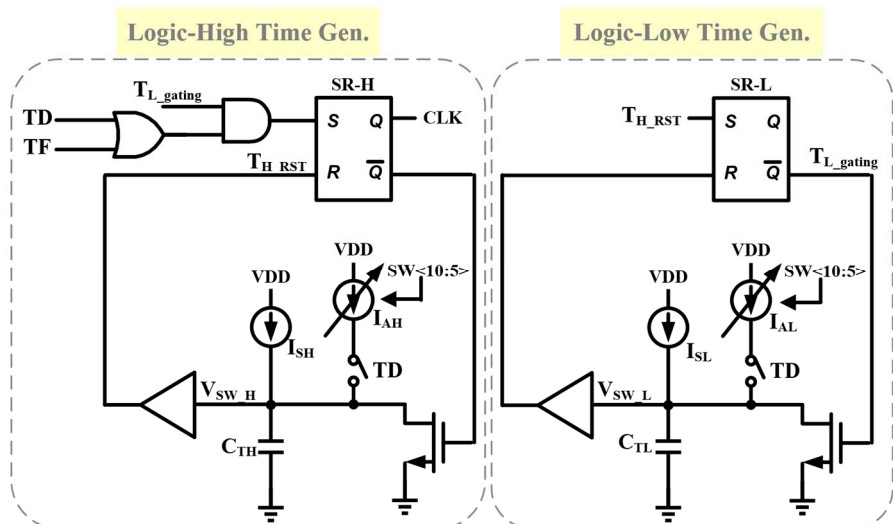

**Figure 9.** Circuit of the proposed event-driven adaptive frequency clock generator (EACG).

As Figure 9 shows, a logic-high time is initiated when the triggered signal TD goes high. The sawtooth voltage $V_{SW\_H}$ starts to rise due to two currents, $I_{SH}$ and $I_{AH}$, charging a capacitor $C_{TH}$. When $V_{SW\_H}$ reaches 1/2 VDD (assume $V_{IN}$ = VDD), the inverter chain output $T_{H\_RST}$ will reset the SR-H latch circuit and set the SR-L latch circuit simultaneously. After SR-L latch is set, a logic-low time is initiated. Like the logic-high time, when $V_{SW\_L}$ reaches 1/2 VDD, the SR-L latch is reset, and $T_{L\_gating}$ will set the SR-H latch. Thus, a toggling clock signal is obtained. After the TD signal is 0 and the TF pulse pulls low, the SR-H latch stops set/reset, and the clock signal will stop toggling. Both logic-high time generator and logic-low time generator have two current sources: a static small current $I_{SH}/I_{SL}$ and a dynamic binary-weighted current array $I_{AH}/I_{AL}$. The dynamic current array will only be connected to the capacitor when the transient event occurs (TD = 1). The dynamic current array can be adapted to the different load currents to achieve adapting frequency for stable operation.

Since the logic-high/low time ($T_H/T_L$) of the EACG is generated by current charging a capacitor, they can be expressed as:

$$T_H = \frac{\frac{1}{2}VDD \cdot C_{TH}}{(I_{SH} + I_{AH})} \tag{5}$$

$$T_L = \frac{\frac{1}{2}VDD \cdot C_{TL}}{(I_{SL} + I_{AL})} \tag{6}$$

We can simplify the power stage of DLDO as an RC circuit, and its time constant can be expressed as:

$$\tau = (R_{ON\_EQ} || R_{LOAD}) \cdot C_{OUT} \tag{7}$$

where $R_{ON\_EQ}$ is the equivalent resistance of the power transistor array, $R_{LOAD}$ is the equivalent resistance of the output load, and $C_{OUT}$ is the output capacitance. It can be seen that the output pole will vary with different load current conditions. In other words, the charging/discharging time of the output node will vary with different load current conditions. In Section 2.3, we cited [13] to give a short discussion about how the output pole and the sampling clock frequency affect the stability of DLDO. To tackle this issue, the proposed EACG adapts the clock frequency with the output current. That is, the clock frequency is designed to be proportional to the output pole frequency. Thus, the phase margin is fixed at various load conditions according to the analysis in reference [13].

Because the DSC switches at the positive clock edge and the $V_{OUT}$ node is sampled at the negative clock edge, the logic-high time $T_H$ is designed to be proportional to the time constant of the power stage, as shown in (8). The proportional coefficient is chosen as more than five, which indicates that the output voltage is fully charged before sampling,

$$T_H \geq 5 \cdot \tau = 5 \cdot (R_{ON\_EQ} || R_{LOAD}) \cdot C_{OUT} \tag{8}$$

Assuming $\beta = V_{OUT}/V_{IN}$, we can derive (9) from the equivalent circuit. Then, $R_{ON\_EQ}$ can be rearranged as (10).

$$\beta = \frac{V_{OUT}}{V_{IN}} = \frac{R_{LOAD}}{R_{ON\_EQ} + R_{LOAD}} \tag{9}$$

$$R_{ON\_EQ} = (\frac{1-\beta}{\beta}) \cdot R_{LOAD} \tag{10}$$

From Equations (5) and (8)–(10), we can obtain the stable design criterion of the charging currents ($I_{SH} + I_{AH}$) of the logic-high time generator versus the load current $I_{LOAD}$ (assume $V_{IN} = VDD$) as (11).

$$I_{SH} + I_{AH} \leq \frac{C_{TH}}{10 \cdot C_{OUT} \cdot (1-\beta)\beta} \cdot R_{LOAD} \tag{11}$$

As can be seen from (8), logic-high time ($T_H$) should be adapted with the load current to ensure $V_{OUT}$ stability. Therefore, the charging current ($I_{SH} + I_{AH}$) of the proposed EACG is designed to be proportional to the load current.

Suppose we just use a fixed high-frequency clock signal for fast transient recovery. In that case, it might violate the criteria of (8) and cause unstable operations, particularly in the light-load condition. Figure 10 shows that if the load current changes from heavy-load to light-load, we still use the high-frequency designed for heavy-load, which might cause the $V_{OUT}$ to start ringing and the system cannot go back to the steady-state. This is because the output node's time constant in the light-load becomes much larger than in the heavy-load, and the clock frequency cannot meet the criteria of (8).

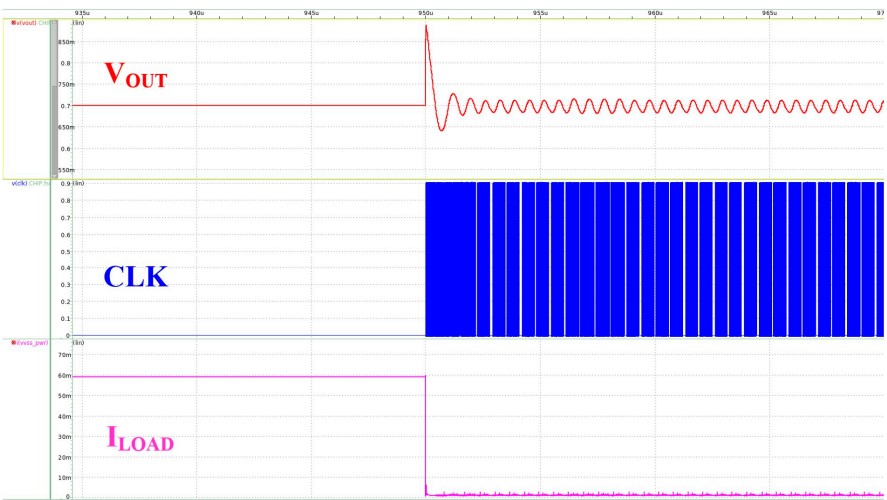

**Figure 10.** The unstable phenomenon in the light-load if the clock frequency cannot meet the criteria of (8).

Moreover, since only the logic-high time needs to fulfill the designed timing criteria of (8), the logic-low time can be shrunk to accelerate the response and reduce the recovery time. In other words, to obtain a shorter recovery time, the duty of CLK could be larger than 50%. The duty of CLK is $C_{TH}/(C_{TH} + C_{TL})$. For example, if $C_{TH}:C_{TL}$ = 3:1, the duty will be 75%.

The load current information can be obtained from the gate signal of the binary-weighted power transistors array, SW<10:0> in Figures 1 and 5 of the proposed DLDO. The current value of SW<10:0> is the turn-on bits of the pMOS array, and it is proportional to the current $I_{LOAD}$. Hence, adaptive charging current $I_{AH}$ is implemented with a binary-weighted current array and pass switches controlled by the TD signal and the gate signal of the DLDO's power transistors array. When the $I_{LOAD}$ is at the lightest condition, the most significant bit (MSB) of turn-on bits is SW<4>. Therefore, the adaptive charging current $I_{AH(AL)}$ is controlled by TD, and the rest bits of SW<10:0> (SW<5>~SW<10>). Considering the power consumption, the adaptive charging current is gated in the steady-state (TD = 0). When the system is in the transient period (TD = 1), the adaptive charging currents provide current $I_{AH(AL)}$ and make ($I_{SH(SL)} + I_{AH(AL)}$) proportional to the present output current.

The DC error is caused by the resolution of the equivalent $R_{ON}$ of the power transistor array. Assume that the system is in steady-state and at the lightest load; the turn-on bit "N" makes $V_{OUT}$ slightly lower than $V_{REF}$, but "N + 1" bit will make $V_{OUT}$ slightly higher than $V_{REF}$. The quantization error between N and N + 1 has to be smaller than the maximum acceptable DC error. For example, if the specification of DC error is designed to be within 10 mV, the DC error value can be expressed as:

$$\begin{cases} V_{OUT} - VDD \cdot \left( \dfrac{R_{LOAD}}{R_{ON_{EQ(N+1)}} + R_{LOAD}} \right) \leq -10 \text{ mV} \\ V_{OUT} - VDD \cdot \left( \dfrac{R_{LOAD}}{R_{ON_{EQ(N)}} + R_{LOAD}} \right) \leq 10 \text{ mV} \end{cases} \tag{12}$$

$$\begin{cases} R_{ON\_EQ(N+1)} = \dfrac{R_{ON}}{N+1} \\ R_{ON\_EQ(N)} = \dfrac{R_{ON}}{N} \end{cases} \tag{13}$$

After some simulations and calculations with (12) and (13), to meet the designed steady-state DC error (<10 mV), the turn-on bits at minimum load current in this work are designed as 5 bits. Because the DC errors are caused by the quantization error of the binary-weighted power transistor array, the quantization error is limited by LSB. The effect of LSB (SW<0>) turned on or off is more significant at light-load conditions. Therefore, the DC errors usually become larger in the light-load condition, as shown in Figure 11.

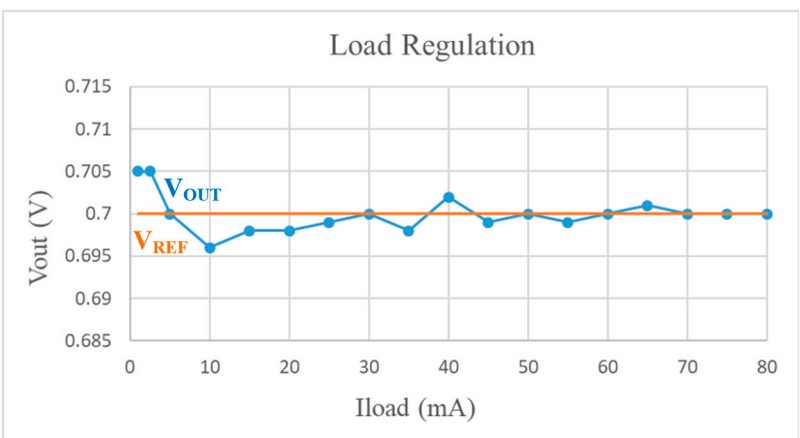

**Figure 11.** Load regulation of the proposed DLDO. (by post-layout simulation, $V_{REF}$ = 0.7 V).

## 4. Measurement Results

The proposed event-driven self-clock DLDO is fabricated in a 40 nm CMOS process. As shown in Figure 12, the chip area of the proposed circuit is 0.0338 mm², which includes the power pMOS array, DSC, EACG, CLTD, the comparator, and other control logic. The remaining chip area is for the 0.3 nF on-chip decoupling capacitor, power rail decoupling capacitors, and the on-chip test load.

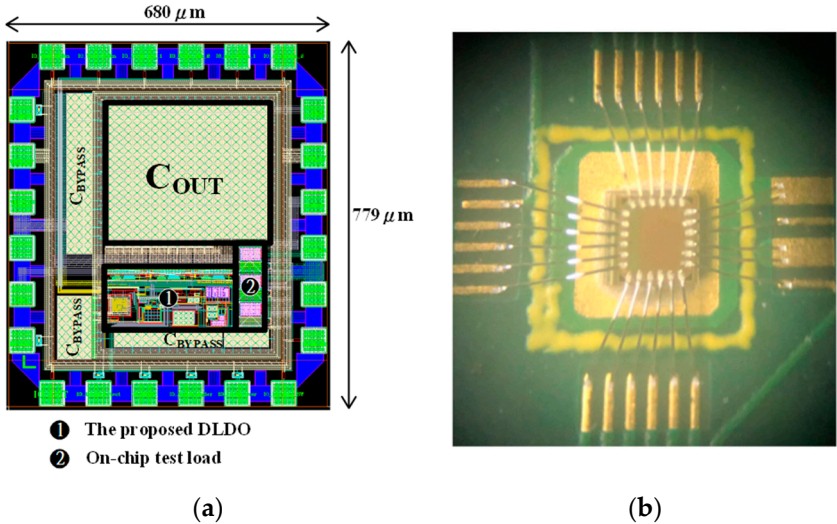

❶ The proposed DLDO
❷ On-chip test load

(**a**)　　　　　　　　　　　　　(**b**)

**Figure 12.** (**a**) Chip layout of the proposed DLDO. (**b**) Die photo of the test chip with bond wires.

Figures 13–15 show the load transient response of the proposed DLDO with different $V_{IN}/V_{OUT}$ and load change conditions to verify the robustness of the proposed DLDO. Figure 13 shows the load transient response with step $I_{LOAD}$ from 1 mA to 50 mA. The load transient is realized by switching the NMOS switches in series with resistors in the on-chip test load. As shown in Figure 13, the self-generated clock signal CLK is only toggling when TD = 1. In other words, the control circuits and the power transistors will not switch when the system is in a steady-state. This implementation can save power and has a cleaner $V_{OUT}$ waveform.

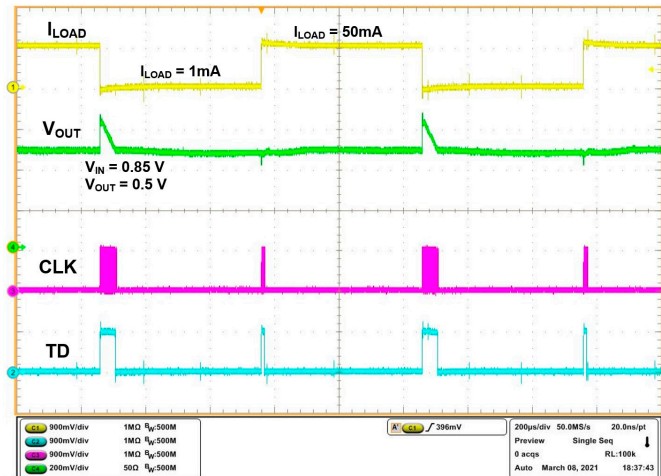

**Figure 13.** Measured load transient response of the proposed DLDO to a periodic square-wave load current (1 mA–50 mA) with $V_{IN}$ = 0.85 V, $V_{OUT}$ = 0.5 V.

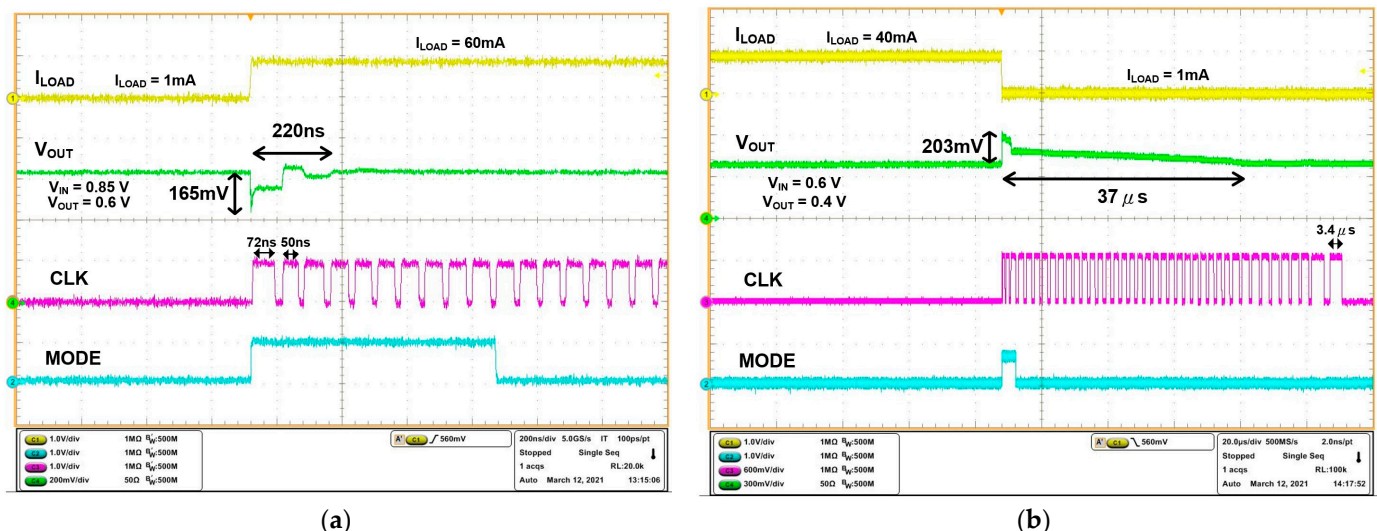

(**a**)　　　　　　　　　　　　　　　　　(**b**)

**Figure 14.** Zoomed-in measured waveform of the load transient period. (**a**) Step-up load transient period. (**b**) Step-down load transient period.

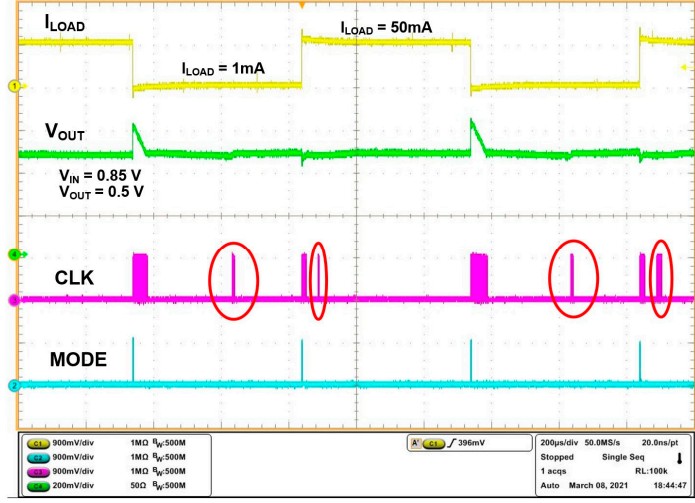

**Figure 15.** Transient waveforms of $V_{OUT}$ clamping by the window of $V_{REF\_L}$ and $V_{REF\_H}$.

Figure 14 shows the zoomed-in oscilloscope captures of the load transient period with varying load current levels. Figure 14a shows the $I_{LOAD}$ step-up load transient period. The proposed DLDO shows a load transient time of 220 ns with 165 mV undershoot. As illustrated in the capture, the DSC used binary search to raise the $V_{OUT}$ immediately at the beginning of MODE = 1, then linear search to make it closer to $V_{REF}$ (MODE = 0). After the DSC turns on the MSB (SW<10>) at the first positive clock edge, the clock frequency is boosted up to match the rising load current to achieve adaptive frequency control. Figure 14a verifies that CLK's logic-high time ($T_H$) keeps adjusting with the binary searching process. In a heavy load condition (60 mA), $T_H \approx 50$ ns, and in the light load condition, $T_H$ is extended up to 3.4 μs. Thus, there is no limit cycle oscillation that happens at light-load or heavy-loads. As illustrated in Figure 14, the duty of the clock signal is larger than 50% as we designed it to reduce the recovery time.

Figure 14b shows the zoomed-in oscilloscope captures of the load transient period with $I_{LOAD}$ from 40 mA to 1 mA. The recovery time of the $I_{LOAD}$ step-down transient is 37 μs, and the overshoot is 203 mV. The adaptive clock frequency is slower when the $I_{LOAD}$ is lower. The DSC uses linear search early when $V_{OUT}$ is close to $V_{REF}$ to avoid unwanted undershoot caused by binary search.

The proposed DLDO can clamp the $V_{OUT}$ within the correct level by the window consisting of $V_{REF\_L}$ and $V_{REF\_H}$. In Figure 15, $V_{REF\_L}$ is enlarged such that the difference between $V_{REF\_L}$ and $V_{REF}$ reduces from the originally designed 10 mV value as in Figure 13. According to the analysis in (12) and (13), the $V_{OUT}$ DC error caused by the mismatch between load current and pass transistor current is large enough to trigger the slow path in CLTD. There are four events (in the red circle) where CLK is triggered, but the MODE signal equals zero. These events show that the $V_{OUT}$ is slightly deviating from $V_{REF}$, and the DSC uses linear search to regulate it back. This method can maintain the $V_{OUT}$ DC errors within the defined level.

Table 2 shows the performance comparison of this work with the state-of-the-art designs. The most common figure-of-merits (FoMs) in (14) [24] and (15) are used to fairly compare the performance considering the transient response and power trade-off.

$$FoM_1 = C_{OUT} \cdot \frac{\Delta V_{DROP}}{(I_{Max} - I_{MIN})} \cdot \frac{I_Q}{(I_{Max} - I_{MIN})} \tag{14}$$

$$FoM_2 = C_{OUT} \cdot \frac{\Delta V_{DROP}}{V_{OUT}} \cdot \frac{I_Q}{(I_{Max} - I_{MIN})} \tag{15}$$

The comparison table shows that the proposed DLDO achieves FoM improvement over the previous measurements in FoM1 and FoM2, which are the only two using adaptive frequency to avoid limit cycle oscillation. With the adaptive frequency self-clocked control method, the proposed DLDO can have a short recovery time and erase the limit cycle oscillation phenomenon without using a complex compensation circuit such as reference [13]. The FoM improvements are owed to the proposed CLTD and EACG. The event-driven asynchronous control means the quiescent current of the system is reduced to a very low level. The CLTD utilizes the power more efficiently than a power-hungry high-speed transient detector. Therefore, this work can achieve a fast transient response without consuming huge power. Furthermore, the quiescent current $I_Q = 26$ μA includes the power consumption of the clock generator EACG, while most of the prior research did not take the power consumption of the power-hungry clock generator into account.

**Table 2.** Performance summary of this work and comparison with state-of-the-art designs.

| Design | [19] JSSC 2017 | [20] JSSC 2017 | [13] JSSC 2018 | [22] ISSCC 2018 | [23] VLSI 2019 | [9] TPE 2022 | This Work |
|---|---|---|---|---|---|---|---|
| Process | 65 nm | 28 nm | 65 nm | 40 nm | 22 nm | 65 nm | 40 nm |
| Control | Event-driven | Time-driven | SAR/PWM | Burst Mode | Event-driven | VCO-based | Event-driven |
| $V_{IN}$ (V) | 0.5–1.0 | 1.1 | 0.5–1 | 0.6–1.1 | 0.55–1.2 | 0.9–1.2 | 0.55–1.0 |
| $V_{OUT}$ (V) | 0.45–0.95 | 0.9 | 0.3–0.45 | 0.5–1 | 0.5–1.15 | 0.5–1.1 | 0.4–0.7 |
| Load Range | 150 μA–500 μA | 4 mA–200 mA | 33.6 μA–2 mA | 1 mA–20 mA | 400 μA–2 A | 150 μA–19 mA | 1 mA–60 mA |
| $C_{OUT}$ (nF) | 0.4 | 23.5 | 0.4 | 4.7 | 7 | 0.2 | 0.3 |
| $I_Q$ (μA) | 12.5 | 110 | 14 | 20 | 2400 | 131 | 26 |
| Sampling clock rate | 200 MHz | N.R. | 1 MHz–240 Hz | 100 MHz | 6 GHz | 500 MHz | External Clock-Less |
| $V_{DROOP}$ @ load step transient test | 22 mV @0.2 mA | 120 mV @180 mA | 40 mV @1.06 mA | 40 mV @19 mA | 100 mV @0.5 A | 78 mV @3 mA | 165 mV @59 mA |
| Recovery time $T_R$ (μs) | 80 | >10 | 0.1 | 1.3 | 0.015 | 0.08 | 0.22 |
| FoM1 (ps) (smaller is better) | 2750 | 9.57 | 199 | 10.4 | 6.7 | 227 | 0.37 |
| FoM2 (pF) (smaller is better) | 0.57 | 1.914 | 0.47 | 0.19 | 3.73 | 0.68 | 0.036 |
| Adaptive frequency for stability | X | X | V | X | X | X | V |

N.R. stands for "data not revealed".

## 5. Conclusions

This paper presents an event-driven self-clocked digital low-dropout regulator with adaptive frequency control. The measurement results based on the prototype chips in the 40 nm CMOS process demonstrate the peak current efficiency of 99.96% at 0.85 V $V_{IN}$ and 0.5 V $V_{OUT}$. We propose a clock frequency adapting technique to improve stability and transient recovery time to 220 ns. The proposed DLDO achieves a fast transient response and low power consumption without any external clocking signal. The total quiescent current is only 26 μA in the steady-state. The comparison table shows that the proposed DLDO is the only work that achieves adaptive frequency for stability without a complex compensation circuit and achieves FoM improvement over the previous works. No external clock and related supply current are required compared to previous work.

**Author Contributions:** Y.-M.C. substantially contributed to the control strategy design, development and implementation of the overall system, analysis and interpretation of the results, and writing and revision of the manuscript. C.-J.C. substantially contributed to the project and research management, review and proofreading of the manuscript. All authors have read and agreed to the published version of the manuscript.

**Funding:** This work was supported in part by Qyi Now Co., Ltd., Taiwan; and in part by the Ministry of Science and Technology, Taiwan.

**Data Availability Statement:** No new data were created or analyzed in this study. Data sharing is not applicable to this article.

**Acknowledgments:** The authors want to thank Taiwan Semiconductor Research Institute (TSRI), Taiwan, and Taiwan Semiconductor Manufacturing Company (TSMC), Taiwan, for chip fabrication support. The authors also want to thank SIMPLIS Technologies Corporation, USA, for providing SIMPLIS simulation tool.

**Conflicts of Interest:** The authors declare no conflict of interest.

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
