# Peer review of "An Event-Driven Self-Clocked Digital Low-Dropout Regulator with Adaptive Frequency Control"

_energies, doi:10.3390/en16124749_

Round 1

Reviewer 1 Report

it presents An Event-Driven Self-Clocked Digital Low-Dropout Regulator with Adaptive Frequency Control, Digital low-dropout (DLDO) is widely used for power management in the system-on-chip 10 (SoC) because of its low-voltage operation and process scalability.It has good theoretical analysis and experimental verification.so i recommend it to be accepted

  •  

null

Author Response

Thanks for your kind comments.

Reviewer 2 Report

In this paper, the authors proposed an event-driven self-clocked digital low-dropout (DLDO) regulator to solve the problem of trade-off between transient response and power consumption of DLDO and clock generator. This paper has excellent academic values, and I recommend to publish it. The English writing of the manuscript is good. My comments are as below.

1. The Figure 5 is not very clear, please change it.

2. Please give more analysis about the Figure 6.

3. In Line 223, “Figure 7 shows the overall operating waveform of the proposed DLDO, taking one 223 severe ILOAD step-up and one smooth ILOAD step-down load transients, for example.”, the description of this sentence is not very clear, please change it.

The English writing of the manuscript is good.

Reviewer 3 Report

The paper presents the design and experimental validation of a circuit for system on chip.

Some remarks:

Eq(1)-(15) Please, be sure that all the symbols in equation were defined in the text.

Fig 4 The second mark pole is the integrator pole? Figure is misunderstanding

Please, be sure that all acronyms were defined in the text.

Line 214 Please, define what is a typical ‘severe load occurred’

Line 228 Please, clarify what are ‘smooth load transient conditions’

Line 256 Please, check the sentence ‘..different load transient situations and make the DSC make the correct response,..’

Please, clarify what is the ‘turn-on bits’

Line 320-321 What is SW<4>, SW<5> SW<10>?

Line 326 Please, clarify the sentence  The lightest load needs to be turned on 5 bits of power transistor array because…’

Line 335-338: please clarify the paragraph

What is typical application of the proposed device?

Round 2

Reviewer 3 Report

the paper was improved